# Human gut *Bacteroides* capture vitamin B$_{12}$ via cell surface-exposed lipoproteins

Aaron G Wexler[1,2], Whitman B Schofield[1,2†], Patrick H Degnan[1,2‡],
Ewa Folta-Stogniew[3], Natasha A Barry[1,2], Andrew L Goodman[1,2]*

[1]Department of Microbial Pathogenesis, Yale University, New Haven, United States; [2]Microbial Sciences Institute, Yale University, New Haven, United States; [3]W.M. Keck Biotechnology Resource Laboratory, Yale University School of Medicine, New Haven, United States

**Abstract** Human gut *Bacteroides* use surface-exposed lipoproteins to bind and metabolize complex polysaccharides. Although vitamins and other nutrients are also essential for commensal fitness, much less is known about how commensal bacteria compete with each other or the host for these critical resources. Unlike in *Escherichia coli*, transport loci for vitamin B$_{12}$ (cobalamin) and other corrinoids in human gut *Bacteroides* are replete with conserved genes encoding proteins whose functions are unknown. Here we report that one of these proteins, BtuG, is a surface-exposed lipoprotein that is essential for efficient B$_{12}$ transport in *B. thetaiotaomicron*. BtuG binds B$_{12}$ with femtomolar affinity and can remove B$_{12}$ from intrinsic factor, a critical B$_{12}$ transport protein in humans. Our studies suggest that *Bacteroides* use surface-exposed lipoproteins not only for capturing polysaccharides, but also to acquire key vitamins in the gut.
DOI: https://doi.org/10.7554/eLife.37138.001

*For correspondence:
andrew.goodman@yale.edu

Present address: †Human and Translational Immunology Program, Yale University School of Medicine, New Haven, United States; ‡Department of Microbiology and Plant Pathology, University of California Riverside, Riverside, United States

Competing interests: The authors declare that no competing interests exist.

## Introduction

Our understanding of the factors that shape gut microbial community composition is largely based on the primary economy of this ecosystem: the flow of carbon from the diet to bacterial biomass and fermentation products. However, an accompanying secondary economy of essential vitamins and other cofactors, which are much less abundant, also plays a critical role in determining bacterial growth rates and resulting microbiome dynamics (*Sonnenburg and Sonnenburg, 2014*). Small molecule cofactor biosynthesis is an energetically costly process and alternate cofactor-independent enzymes can be less efficient (*Roth et al., 1993*), favoring microbes that can best acquire these nutrients from their environment.

The complex organometallic cofactor vitamin B$_{12}$ (cobalamin) is representative of this challenge: de novo biosynthesis requires the coordinated activity of nearly 30 dedicated enzymes (*Roth et al., 1993*). Although bacteria have evolved cofactor-independent alternatives to many B$_{12}$-dependent enzymes (e.g. B$_{12}$-independent methionine synthase MetE and ribonucleotide reductase NrdEF), even species that lack vitamin biosynthetic machinery maintain their B$_{12}$-dependent enzymes (e.g., methionine synthase MetH and ribonucleotide reductase NrdZ) for use when vitamin B$_{12}$ is available in the environment (*Degnan et al., 2014a*; *Degnan et al., 2014b*; *Young et al., 2015*). In the human gut, most species encode B$_{12}$-dependent enzymes. The great majority of these species also encode transport systems for capturing B$_{12}$ from the environment, either instead or in addition to vitamin biosynthetic pathways (*Degnan et al., 2014a*). Although bacteria utilize many vitamin B$_{12}$-like molecules (corrinoids), transport and utilization proteins and regulatory elements are typically referred to as B$_{12}$-dependent, in keeping with their initial characterization.

The machinery used by Gram-negative bacteria to transport vitamin B$_{12}$ and other corrinoids has been studied extensively in *Escherichia coli* and has been used as a model for TonB-dependent

**eLife digest** Eating is the first step in an hours-long process that extracts the nutrients we need to live. It not only nourishes us, but also a vast community of bacteria in our gut called the microbiota. The gut microbiota acts like an extension of our immune system and helps us stay healthy in many ways. For example, it blocks pathogens from making us sick. But too many gut bacteria in the wrong parts of our intestines can be harmful.

Some people are prone to developing a dangerous overgrowth of bacteria in their small intestine where most of our dietary nutrients get absorbed. This overgrowth can lead to many problems including vitamin B12 deficiency even when they eat plenty of it. To understand why, scientists must learn how microbes affect our ability to absorb nutrients from food and how the microbes themselves capture nutrients like vitamin B12 as they pass through our digestive tract.

Now, Wexler et al. show that some gut microbes may be able to pirate vitamin B12 from us as it passes through the digestive tract. Wexler et al. showed that a protein called BtuG on the surface of a type of gut bacteria called *Bacteriodes* grabs onto vitamin B12 with extraordinary strength. In fact, these bacterial proteins bind to vitamin B12 so strongly that they can even pry it away from our own vitamin B12 collecting protein.

When *Bacteriodes* with and without BtuG were placed in mice with no gut bacteria of their own, bacteria with BtuG rapidly outcompeted those lacking the protein. The experiments suggest that competition for vitamin B12 among microbes has favored bacteria that are better at capturing the nutrient. More studies are needed to learn whether BtuG contributes to vitamin B12 deficiencies in humans with gut bacteria overgrowth and determine the best ways to combat such deficiencies.
DOI: https://doi.org/10.7554/eLife.37138.002

transport. In *E. coli*, extracellular $B_{12}$ is transported into the periplasm through the outer membrane β-barrel protein BtuB in a TonB-dependent manner (*Bassford et al., 1976*; *Bassford and Kadner, 1977*). The periplasmic protein BtuF subsequently binds to and delivers $B_{12}$ to the ABC-type transporter BtuCD in the inner membrane, which brings the vitamin into the cytoplasm (*Cadieux et al., 2002*).

In previous studies, we established that the most abundant Gram-negative bacteria in the human gut (Bacteroidetes) encode a diverse array of $B_{12}$ transport systems in $B_{12}$-riboswitch regulated loci, often with multiple locus architectures per genome (*Degnan et al., 2014a*). *Bacteroides thetaioatomicron* serves as a model for defining the role of these transporters, as it does not encode $B_{12}$ biosynthetic machinery, and its repertoire of $B_{12}$ transporters determine in vivo fitness in a diet- and community context-dependent manner (*Degnan et al., 2014a*; *Goodman et al., 2009*). In the course of these studies, we noticed that *B. thetaiotaomicron* and other human gut *Bacteroides* also maintain a heterogeneous repertoire of additional genes, nearly all of unknown function, in these $B_{12}$ transport loci. *E. coli* also encodes additional genes within its $B_{12}$ transport operons (e.g., *btuE* is positioned between *btuC* and *btuD*), but these do not play a role in $B_{12}$ transport (*de Veaux et al., 1986*; *Arenas et al., 2010*).

Here we report that unlike BtuE in *E. coli*, *Bacteroides* accessory proteins can play a critical role in $B_{12}$ transport. Using the *B. thetaiotaomicron* homolog of a highly conserved accessory protein (BT1954; hereafter called BtuG2) as a representative, we establish that BtuG2 localizes to the cell surface, interacts with BtuB, and determines the ability of *B. thetaiotaomicron* to transport $B_{12}$ and persist in the mammalian gut. Furthermore, BtuG2 directly binds vitamin $B_{12}$ with femtomolar affinity, thereby enabling it to acquire this vitamin from intrinsic factor, a critical $B_{12}$ transport protein in humans.

## Results

### Bacteroidetes *btuG* homologs display widespread genetic linkage to vitamin $B_{12}$ transport genes and facilitate cyanocobalamin acquisition

Nearly all of the *Bacteroides* $B_{12}$ transport loci include homologs of a hypothetical gene that is exclusively found in the Bacteroidetes phylum (*Figure 1A*). We refer to these homologs as BtuG; one of

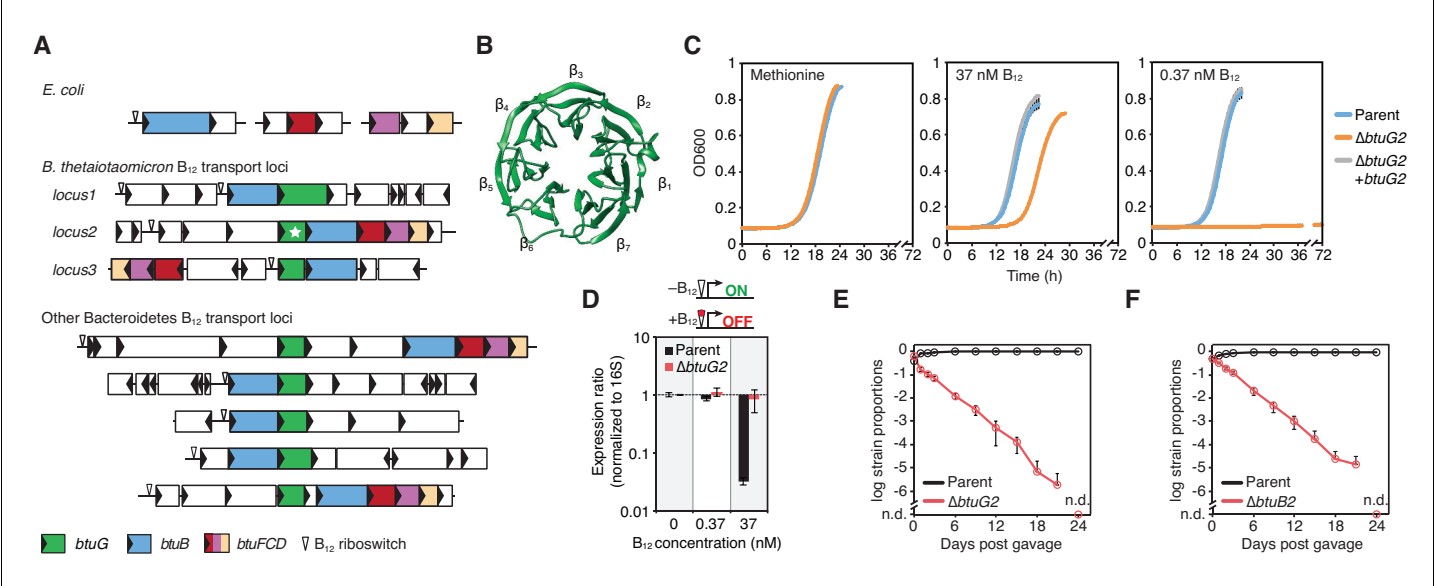

**Figure 1.** BtuG homologs are exclusively found among the Bacteroidetes, facilitate the acquisition of cyanocobalamin in vitro and confer a fitness advantage in gnotobiotic mice. (**A**) Genetic loci encoding corrinoid transport components in *E. coli*, *B. thetaiotaomicron* and other Bacteroidetes. (**B**) BtuG2 (PDB 3DSM) adopts a seven-bladed β-propeller fold. (**C**) Growth curves for the *B. thetaiotaomicron* parent strain, *btuG2* deletion strain or complemented strain grown in minimal media supplemented with methionine or vitamin $B_{12}$. Data are representative of three independent trials; error bars indicate ±SD from three technical replicates. (**D**) Gene expression ratios for *B. thetaiotaomicron* strains grown in minimal media with methionine and indicated concentrations of vitamin $B_{12}$. Expression of *locus2* (BT1956) was normalized first to 16S rRNA and then to each strain's expression in 0 nM vitamin $B_{12}$. Data are representative of two independent trials; error bars indicate ± SD from three biological replicates. (**E, F**) *B. thetaiotaomicron* strain ratios determined from gDNA extracted from fecal samples collected over time from gnotobiotic mice. (n.d., not detected; *n* = 4 mice/group; error bars indicate ± SD).

DOI: https://doi.org/10.7554/eLife.37138.003

The following figure supplement is available for figure 1:

**Figure supplement 1.** A BtuG homolog is required for *B. thetaiotaomicron* fitness in gnotobiotic mice.

DOI: https://doi.org/10.7554/eLife.37138.004

which (BtuG2 from *B. thetaiotaomicron*, marked with a star in *Figure 1A*) has been crystallized and adopts a seven-bladed β-propeller fold (PDB 3DSM; *Figure 1B*). *B. thetaiotaomicron* encodes three genetic loci with vitamin $B_{12}$ transport genes (*locus1*, *locus2* and *locus3*; *Figure 1A*). Each locus encodes a *btuG* homolog (*btuG1*, *btuG2* and *btuG3*, respectively) adjacent to a homolog of *btuB* (*Figure 1A*). Using BtuG1, BtuG2 and BtuG3 from *B. thetaiotaomicron* as representatives, we identified 112 putative BtuG homologs in 313 genome-sequenced human gut bacterial strains (*Degnan et al., 2014a*) by an initial BlastP search (*Supplementary file 1*). 106 are encoded in 106 of 109 *btuB*-containing operons identified previously (*Degnan et al., 2014a*). The six remaining *btuG* homologs identified by BlastP have operon annotations affected by incomplete genome assemblies, however, five are associated with a corrinoid riboswitch and one or more *btu* transport genes. Two of the three remaining *btuB*-containing operons encode a divergent *btuG* gene (e-value >1e-10, but Phyre2 match to the BtuG2 crystal structure). No homologs were detected outside of the Bacteroidetes using the defined BlastP parameters (*Supplementary file 1*).

We sought to test whether BtuG is needed for $B_{12}$-dependent growth, given the near universal genetic linkage of *btuG* homologs to known $B_{12}$ transport genes. Because *B. thetaiotaomicron* encodes three homologous $B_{12}$ transport loci (and corresponding *btuG* genes) that complicate the ability to assign functions to specific genes, we established a simplified genetic background that lacks *locus1* and *locus3* (hereafter referred to as the *B. thetaiotaomicron* 'parent' strain). We created an in-frame, unmarked deletion of *btuG2* in this parent strain and tested its ability to grow in minimal media supplemented with cyanocobalamin ($B_{12}$-dependent growth; *B. thetaiotaomicron* encodes MetH but not MetE) or methionine ($B_{12}$-independent growth). We chose cyanocobalamin concentrations of 37 nM and 0.37 nM because these concentrations repress (37 nM) or activate (0.37 nM) $B_{12}$

riboswitches in this species (*Sonnenburg et al., 2005*; *Martens et al., 2008*; *Degnan et al., 2014a*). As compared with its growth in methionine medium, the *btuG2* deletion strain had a ~4 hr longer lag phase in 37 nM cyanocobalamin medium and did not display any growth during 72 hr in 0.37 nM cyanocobalamin medium (*Figure 1C*). By contrast, the parent and complemented strains grew indistinguishably in all three media.

Vitamin $B_{12}$ riboswitches are RNA aptamers that bind $B_{12}$ directly and repress downstream gene expression, providing a biosensor for intracellular levels of this cofactor (*Fowler et al., 2010*). To determine the contribution of BtuG2 to intracellular corrinoid accumulation, we used the $B_{12}$ riboswitch of *locus2* as a biosensor. Quantification of gene expression by qRT-PCR revealed that the parent strain represses riboswitch-dependent gene expression ~70 fold in culture medium containing 37 nM cyanocobalamin as compared to medium with 0.37 nM and 0 nM cyanocobalamin. By contrast, the *btuG2* deletion strain fails to repress $B_{12}$ riboswitch-regulated gene expression in any concentration of extracellular cyanocobalamin (*Figure 1D*). These data suggest that BtuG2 contributes significantly to $B_{12}$ accumulation within *B. thetaiotaomicron* cells.

$B_{12}$ transport machinery encoded in *locus2* are critical for *B. thetaiotaomicron* fitness in gnotobiotic mice (*Goodman et al., 2009*). To compare the relative contribution of BtuG2 and BtuB2 to fitness in the gut, we colonized germfree mice with a 1:1 mixture of the parent strain and Δ*btuG2* (*Figure 1E* and *Figure 1—figure supplement 1*) or Δ*btuB2* (*Figure 1F*), and monitored the relative abundance of each strain in fecal samples collected over time. In both groups of mice, the parent strain dominated while the abundance of the mutant strain dropped continuously until it was no longer detected on day 24 (*Figure 1E–F*). Surprisingly, these data indicate that the absence of BtuG2 has a similarly deleterious impact on in vivo fitness as the absence of the outer membrane transporter BtuB2. Together, these studies suggest that BtuG plays a critical role in mediating *B. thetaiotaomicron* $B_{12}$ transport in the gut.

## BtuG2 is a lipoprotein that localizes to the cell surface and associates with $B_{12}$ transport machinery

We next sought to determine the subcellular localization of BtuG2 to better understand its role in $B_{12}$ transport. Aligning the first 90 amino acids of 114 homologs of BtuG (*Supplementary file 1*) using ClustalW (*Larkin et al., 2007*) and displaying their conservation as a sequence logo (*Crooks et al., 2004*; *Schneider and Stephens, 1990*) revealed a number of important clues: firstly, nearly all homologs have a conserved cysteine residue within their first ~18–40 amino acids (Cys-32 of BtuG2) (*Inouye et al., 1983*); secondly, this cysteine is preceded by a conserved lipobox-like sequence typical of lipidated proteins (VFGS in BtuG2) (*von Heijne, 1989*); thirdly, this cysteine is followed by a conserved lipoprotein export signal or LES (MKWD in BtuG2), a feature exclusive to Bacteroidetes that allows lipoproteins to be flipped from the inner to the outer leaflet of the outer membrane (*Figure 2A*) (*Lauber et al., 2016*). Thus, from their primary sequences alone, we predicted BtuG homologs to be surface-exposed lipoproteins.

To test whether BtuG2 is indeed surface-exposed, we treated intact cells grown in minimal medium with methionine (but without cyanocobalamin) with varying concentrations of proteinase K (0–100 μg/mL), ran whole cell lysates on an SDS-PAGE gel, and probed by Western blot for BtuG2 or a periplasmically localized control protein SusA (BT3704) appended with a C-terminal HA-tag (*Shipman et al., 1999*). While SusA was protected from protease treatment even at the highest concentration of protease, BtuG2 was progressively degraded at increasing concentrations of protease, consistent with it being surface-exposed (*Figure 2B*). Moreover, BtuG2 associated most strongly with the membrane of fractionated *B. thetaiotaomicron* cells (parent strain), and was even found in the supernatant fraction, consistent with its localization to the interface of the outer membrane of the cell and the extracellular milieu (*Figure 2C*).

Alteration of these surface-localizing sequence features diminished BtuG2 stability, complicating efforts to directly assess the contribution of these sequences to protein localization (*Figure 2—figure supplement 1A*). Deleting the putative signal sequence (aa 2 – 31), mutating Cys-32 to alanine, or changing three residues of the LES (Lys-34, Trp-35 and Asp-36) to alanine all resulted in a lack of BtuG2 detection in whole cell lysates by Western blot, despite normal levels of transcription (*Figure 2—figure supplement 1A–B*). By contrast, replacing the putative signal sequence with aa 1 – 18 of the known surface-exposed lipoprotein SusD (BT1762; aa 1 – 18) (*Shipman et al., 2000*; *Glenwright et al., 2017*), or replacing the signal sequence and LES (BtuG2 aa 1 – 37) with the signal

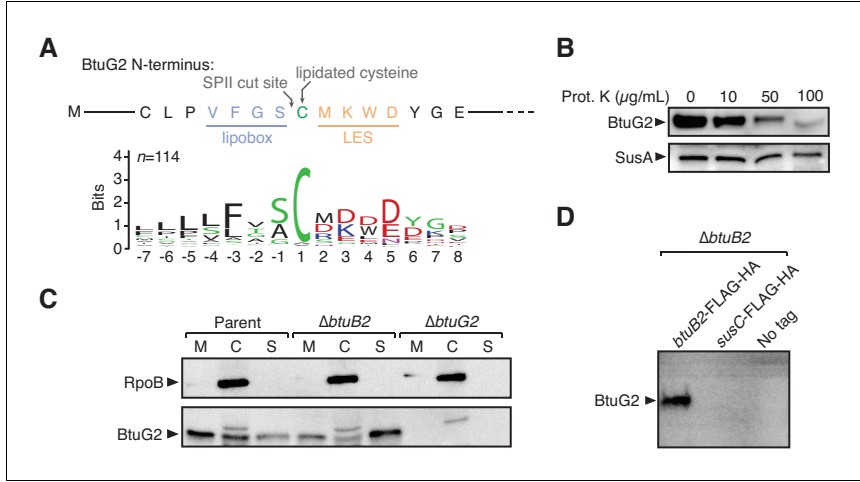

**Figure 2.** BtuG2 is a cell surface-exposed lipoprotein that interacts with the outer membrane transporter BtuB2. (A) N-terminus of BtuG2 and sequence logo of 114 BtuG homologs reveal sequence signatures indicative of a surface-exposed lipoprotein, including a lipobox, adjacent cysteine residue and lipoprotein export signal (LES). (B) Protease degradation of BtuG2 on whole *B. thetaiotaomicron* cells suggests BtuG2 is surface-exposed. SusA is a periplasmic control. Data are representative of three independent trials. (C) *B. thetaiotaomicron* cells separated into membrane (M), cytoplasm/periplasm (C), and supernatant (S) fractions reveal that BtuG2 is predominantly associated with the membrane in parent cells, but predominantly associated with the supernatant in Δ*btuB2* cells. Data are representative of four independent trials. (D) In vivo pull-down of BtuG2 by TAP-tagged BtuB2 suggests an interaction with BtuB2, but not with an unrelated outer membrane β-barrel protein SusC. Data are representative of two independent trials.
DOI: https://doi.org/10.7554/eLife.37138.005
The following figure supplement is available for figure 2:

**Figure supplement 1.** BtuG2 lipoprotein sequence signatures are required for protein production and can be functionally replaced with N-terminal sequences from the unrelated cell surface lipoprotein SusD.
DOI: https://doi.org/10.7554/eLife.37138.006

sequence and LES of SusD (aa 1 – 24) complemented a *btuG2* deletion strain both in terms of protein production and $B_{12}$-dependent growth (*Figure 2—figure supplement 1C–D*). These results indicate that the N-terminal residues of BtuG2 are critical for protein stability, and that these residues can be functionally replaced with the corresponding sequences from the surface-exposed lipoprotein SusD.

Given that BtuG2 exhibits sequence signatures and protease sensitivity indicative of a surface-exposed lipoprotein, is localized to the cell membrane, and contributes to $B_{12}$ acquisition, we hypothesized that it might interact with the outer membrane $B_{12}$ transporter BtuB2. Consistent with this hypothesis, we found that deletion of *btuB2* changes the predominant localization of BtuG2 from the membrane fraction to the culture supernatant (*Figure 2C*). Furthermore, tandem affinity purification (TAP) of BtuB2-associated proteins in growing *B. thetaiotaomicron* cells readily pulls down BtuG2 (*Figure 2D*). By contrast, TAP of an untagged strain, or of a strain with the TAP epitopes appended to an unrelated TonB-dependent outer membrane β-barrel-type transporter (SusC; BT1763) fails to pull down BtuG2. These data suggest that BtuG2 associates with known $B_{12}$ transport machinery.

## BtuG2 directly binds cyanocobalamin with femtomolar affinity

All components of the canonical vitamin $B_{12}$ transport pathway—BtuB, BtuF and BtuCD—bind cyanocobalamin directly during the process of transport from outside the cell into the periplasm and ultimately the cytoplasm. To test whether BtuG2 also binds cyanocobalamin, we expressed and purified BtuG2-10xHis in *E. coli*. We then took advantage of the ability of cyanocobalamin to absorb light at a wavelength of 362 nm, and aromatic amino acids within BtuG2 to absorb light at 280 nm, by performing size-exclusion chromatography with multi-angle light scattering (SEC-MALS) on BtuG2 after co-incubation with equimolar cyanocobalamin. At both wavelengths, we observed nearly

identical traces corresponding to the elution volume for BtuG2, indicating that BtuG2 can directly bind cyanocobalamin in vitro (*Figure 3A*). Purified, recombinant BtuG homologs from *B. vulgatus*, *B. uniformis* and *B. coprophilus* (BVU2056, BACUNI04578 and BACCOPRO02032, respectively) also exhibit this function (*Figure 3—figure supplement 1*).

We then sought to determine the kinetics and affinity of cyanocobalamin binding by BtuG2. Because BtuG2-cyanocobalamin saturation occurs too rapidly to measure the dissociation constant accurately by isothermal titration calorimetry (data not shown), we used surface plasmon resonance (SPR) to determine a $K_D$ of $1.87 \pm 0.76 \times 10^{-13}$ M for BtuG2-cyanocobalamin binding and $1.93 \pm 0.63 \times 10^{-13}$ M for BtuG2-dicyanocobinamide binding (*Figure 3B*). Both ligands bind BtuG2 at a 1:1 ratio. BtuG2 binds to cyanocobalamin with a measured $k_{on} = 1.40 \pm 0.05 \times 10^9$ M$^{-1}$s$^{-1}$ and a $k_{off} = 2.59 \pm 0.96 \times 10^{-4}$ s$^{-1}$; similarly, BtuG2 binds to dicyanocobinamide with a measured $k_{on} = 2.61 \pm 1.56 \times 10^9$ M$^{-1}$s$^{-1}$ and a $k_{off} = 4.54 \pm 1.38 \times 10^{-4}$ s$^{-1}$. Together, these measurements establish that BtuG2 binds cyanocobalamin and a corrinoid precursor with femtomolar affinity, at a rate generally observed for diffusion-limited enzymes and proteins. This ligand interaction is maintained for over 1 hr on average before spontaneous dissociation (*Corzo, 2006*). Notably, surface electrostatic analysis of the crystal structure of BtuG2 reveals a predominantly positive electrostatic potential on the face of BtuG2 displaying the C-terminal 6x-His tag, and a predominantly negative electrostatic potential on the opposing face (*Figure 3—figure supplement 2A*). The coordinated cobalt ion of a corrinoid carries a positive charge ranging from +1 to +3, depending in part on the upper ligand (–CN, –Me, –Ado, –OH) (*Obeid et al., 2015*). Therefore, if surface electrostatic charges on BtuG2 are involved in orienting corrinoids to facilitate protein-ligand interactions, these forces should draw the ligand into the negative electrostatic face of BtuG2.

## BtuG2 can function extracellularly as an early step in vitamin B$_{12}$ acquisition and confers a fitness advantage to producer cells

Because BtuG2 modulates intracellular corrinoid levels, is surface-exposed, associates with BtuB2 and binds cyanocobalamin directly, we reasoned that it might act as a critical extracellular step in the process of capturing and transporting corrinoids. To test whether BtuG2 can function extracellularly, we measured growth of *btuG2* mutant cells upon supplementation with supernatants from *btuB2* mutant cultures (this strain disproportionally partitions BtuG2 to the supernatant; *Figure 2C*).

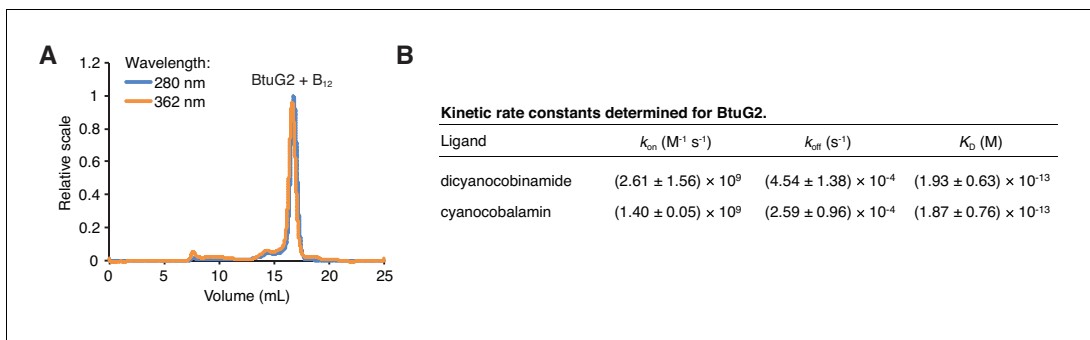

**Figure 3.** BtuG2 binds cyanocobalamin and its corrinoid precursor with femtomolar affinity. (**A**) SEC-MALS traces for recombinant BtuG2 incubated with vitamin B$_{12}$. BtuG2 (protein) absorbance is measured at 280 nm; vitamin B$_{12}$ absorbance is measured at 362 nm. Data are representative of three independent trials. (**B**) Kinetic rate constants and equilibrium dissociation constant for BtuG2 binding to dicyanocobinamide and cyanocobalamin determined by SPR. Data are representative of three independent trials; error represents ± SD from rate constants measured across three Biacore chip cells.

DOI: https://doi.org/10.7554/eLife.37138.007

The following figure supplements are available for figure 3:

**Figure supplement 1.** Cyanocobalamin binding by diverse BtuG homologs.
DOI: https://doi.org/10.7554/eLife.37138.008

**Figure supplement 2.** Surface electrostatic profiles of the seven-bladed β-propeller proteins BtuG2, EUBREC_1955 and MSMAS_RS11935, and the globular enzyme acetylcholinesterase.
DOI: https://doi.org/10.7554/eLife.37138.009

To this end, we collected, filter-sterilized, ultra-centrifuged, and concentrated culture supernatants from *B. thetaiotaomicron* Δ*locus1* Δ*locus3* Δ*btuB2* (or Δ*locus1* Δ*locus3* Δ*btuG2* as a control) strains grown to exponential phase in minimal medium lacking cyanocobalamin and supplemented with methionine. Concentrated supernatants were incubated in the presence or absence of 0.37 μM cyanocobalamin, diluted and concentrated repeatedly to remove unbound ligand and residual methionine, and introduced to recipient cells in medium lacking both cyanocobalamin and methionine.

Under these conditions, Δ*btuG2* recipient cells grow robustly when provided with BtuG2-containing culture supernatant that had been incubated with cyanocobalamin (*Figure 4A*). By contrast, the same recipient strain failed to grow when provided culture supernatant from a Δ*btuG2* strain incubated with cyanocobalamin, or when provided BtuG2-containing culture supernatant incubated with PBS instead of cyanocobalamin (*Figure 4A*). Further, BtuG2-containing culture supernatant incubated with cyanocobalamin failed to rescue growth of recipient cells lacking corrinoid transport machinery. These data suggest that BtuG2 can function from the exterior of cells in trans to promote cyanocobalamin-dependent growth. Notably, BtuG2-6xHis purified from *E. coli* failed to restore cyanocoblamin-dependent growth to a Δ*btuG2* mutant, suggesting that lipidation or some other *Bacteriodes*-specific feature may be important for protein function (data not shown).

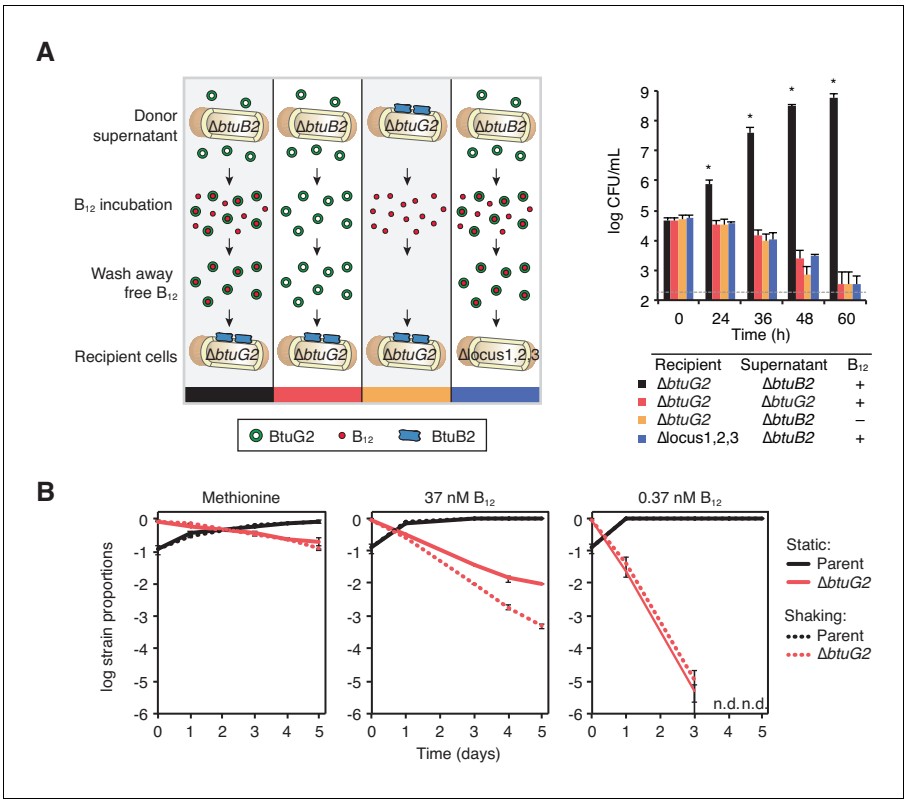

**Figure 4.** BtuG2 can function extracellularly as a corrinoid delivery protein and confers a fitness advantage to BtuG2 producer cells. (**A**) Schematic (left) and measured CFUs (right) from an experiment in which *B. thetaiotaomicron* recipient cultures (Δ*btuG2* or Δ*locus1,2,3*) received donor supernatant from *B. thetaiotaomicron* donor strains (Δ*btuB2* or Δ*btuG2*), with or without vitamin $B_{12}$. Recipient cultures were plated for CFUs over time. Data are representative of four independent trials; *$p<0.05$ for black bars compared against red, yellow or blue bars; error bars indicate ± SD from two biological replicates. (**B**) *B. thetaiotaomicron* parent and Δ*btuG2* strains were co-cultured in minimal media supplemented with methionine or the indicated concentrations of vitamin $B_{12}$ and incubated at 37 °C anaerobically either statically (solid lines) or shaking (dotted lines). Cells were passaged into fresh media daily and strain abundances were determined by qRT-PCR using barcode-specific primers. n.d., not detected; data are representative of two independent trials; error bars indicate ± SD from three technical replicates.

DOI: https://doi.org/10.7554/eLife.37138.010

The observation that BtuG2 can function in trans raises the possibility that this protein could act as a public good, secreted into the environment and shared across cells in the population. However, BtuG2-producing parent cells fail to rescue the cyanocobalamin-dependent growth defect of a ΔbtuG2 strain when the two strains are co-cultured (*Figure 4B*), suggesting that this protein primarily functions in cis. This is consistent with its localization as a membrane-bound lipoprotein (*Figure 2*) and the competitive defect of a ΔbtuG2 strain in the presence of parent *B. thetaiotaomicron* in the mouse gut (*Figure 1E* and *Figure 1—figure supplement 1*).

## *B. thetaiotaomicron* can use BtuG2 to acquire cyanocobalamin from human intrinsic factor

Because BtuG2 is surface-exposed and binds cobalamin with femtomolar affinity, we wondered if it might affect the function of human $B_{12}$-binding proteins that transport this vitamin through the length of the gastrointestinal tract. Humans absorb cobalamin from their diet with the help of two carrier proteins. The first, haptocorrin, is secreted from salivary glands and binds the vitamin as it is released from food broken down in the stomach; the second, intrinsic factor (IF), is released from parietal cells in the stomach and binds cobalamin in the duodenum following the degradation of haptocorrin by host enzymes (*Nielsen et al., 2012*). IF then carries the vitamin through several meters of intestinal tract to the distal ileum, where receptors on intestinal epithelial cells allow for the uptake of IF and its vitamin cargo (*Nielsen et al., 2012*). As IF traverses the small intestinal lumen, it encounters increasing densities of gut microbes (from $\sim 10^3$ to $\sim 10^8$ cells/g) (*Scheithauer et al., 2016*).

To test whether *B. thetaiotaomicron* can use BtuG2 to acquire vitamin $B_{12}$ from IF, we first incubated recombinant human IF with cyanocobalamin, diluted and concentrated repeatedly to remove unbound ligand, provided the IF-cyanocobalamin complexes to *B. thetaiotaomicron* cells in minimal media without exogenous cyanocobalamin or methionine, and measured culture growth over time. The parent *B. thetaiotaomicron* strain grew readily when provided IF-cyanocobalamin (*Figure 5A*). By contrast, IF alone was not sufficient to allow bacterial growth, and recipient cells lacking *btuG2* do not grow when provided IF-cyanocobalamin. Further, the parent strain is unable to grow when provided the filtrate from the last IF-cobalamin wash step, indicating that an insignificant amount of cobalamin dissociates from IF during dilution and concentration (*Figure 5A*). These results indicate that *btuG2*-encoding *B. thetaiotaomicron* cells are capable of acquiring cyanocobalamin from IF.

To test whether BtuG2 acquires cobalamin from IF directly, we incubated cyanocobalamin with IF and/or recombinant BtuG2 and determined the relative amounts of the vitamin associated with each protein by SEC-MALS. As expected, incubation of BtuG2 with cyanocobalamin produces a distinct absorbance peak at 362 nm corresponding to the elution volume for BtuG2 (*Figure 5B*). Similarly, incubation of IF with cyanocobalamin produces a 362 nm absorbance peak at the elution volume corresponding to IF. Notably, addition of BtuG2 to IF-cobalamin shifts the majority of the 362 nm absorbance to the elution volume for BtuG2 (*Figure 5B*). This suggests that BtuG2 can directly acquire cobalamin from an IF-cobalamin complex.

We next sought to determine whether this direct transfer of cobalamin from IF to BtuG2 allows *B. thetaiotaomicron* to grow on cobalamin acquired from IF-cobalamin complexes. Indeed, ΔbtuG2 recipient cells grow readily when provided ΔbtuB2 culture supernatants supplemented with IF-cobalamin complexes (*Figure 5C*). The same recipient cells do not grow when provided ΔbtuG2 culture supernatants supplemented with IF-cobalamin complexes, when provided ΔbtuB2 culture supernatants supplemented with IF alone, or when provided ΔbtuB2 culture supernatants supplemented with the filtrate from the last wash of IF-cobalamin complexes (*Figure 5C*). Collectively, these results suggest that *B. thetaiotaomicron* can use BtuG2 to acquire cobalamin that is already bound to the host protein responsible for transporting this vitamin through the gastrointestinal tract.

## Discussion

Early studies documenting increased vitamin requirements for germfree animals suggested that the microbiota plays a critical role in contributing these essential cofactors to the host (*Barnes, 1967*). In the case of vitamin $B_{12}$, however, it is unlikely that the microbiota makes a significant contribution to host vitamin supply. Indeed, $B_{12}$ constitutes less than 2% of fecal corrinoid pools in humans, and supplementation studies suggest that gut microbes efficiently convert dietary $B_{12}$ into alternate

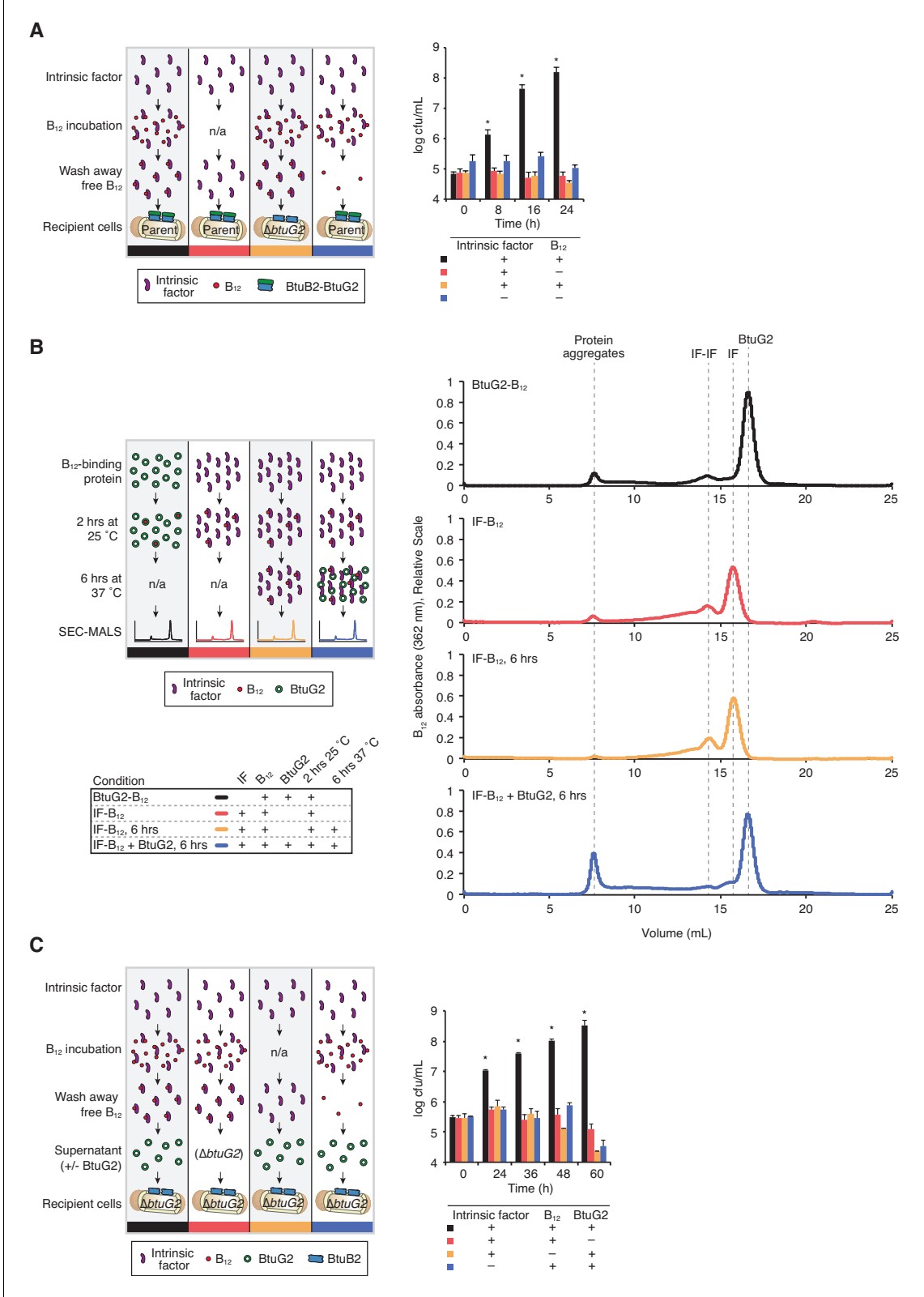

**Figure 5.** BtuG2 mediates vitamin B$_{12}$ piracy from host intrinsic factor. (**A**) Schematic (left) and measured CFUs (right) from an experiment in which *B. thetaiotaomicron* recipient cultures (parent or Δ*btuG2*) received recombinant human IF with or without vitamin B$_{12}$, or the filtrate from the last IF–B$_{12}$ wash. Recipient cultures were plated for CFUs over time. Data are representative of two independent trials; *p<0.05 for black bars compared against red, yellow or blue bars; error bars indicate ± SD from two biological replicates. (**B**) Schematic (left) and SEC-MALS traces at 362 nm absorbance (right)

*Figure 5 continued on next page*

*Figure 5 continued*

of recombinant human IF and/or recombinant BtuG2 incubated with vitamin B$_{12}$. 362 nm absorbance measures B$_{12}$-associated proteins and each trace represents one of the four conditions illustrated in the schematic. Data are representative of two independent trials. (**C**) Schematic (left) and CFUs (right) from an experiment in which *B. thetaiotaomicron* recipient cultures (ΔbtuG2) received recombinant human IF with or without vitamin B$_{12}$ or the filtrate from the last IF–B$_{12}$ wash, and donor supernatant from *B. thetaiotaomicron* strains (ΔbtuB2 or ΔbtuG2). Recipient cultures were plated for CFUs over time. Data are representative of two independent trials; *p<0.05 for black bars compared against red, yellow or blue bars; error bars indicate ± SD from two biological replicates.

DOI: https://doi.org/10.7554/eLife.37138.011

corrinoids that cannot be used by humans (*Allen and Stabler, 2008*). In this report, we describe a novel microbial factor, BtuG, that could pose a more direct obstacle to the host's absorption of this essential vitamin.

What evolutionary forces drove the Bacteroidetes, unlike other Gram-negative phyla, to incorporate an additional component into their B$_{12}$ transport pathway that binds B$_{12}$ with such high affinity? One possible answer lies in the gut environment, where bacteria co-exist at densities of 10$^{11}$ cells per gram or higher (*Whitman et al., 1998*). Under these conditions, adaptations that increase corrinoid capture could allow cells to minimize their requirement for energetically costly vitamin biosynthetic pathways. Indeed, many human gut Bacteroidetes encode incomplete vitamin B$_{12}$ biosynthesis pathways; *B. thetaiotaomicron* is missing this pathway entirely. Selection for increased corrinoid binding affinity in BtuG could conversely permit mutations that decrease the ability of BtuB to directly capture these molecules from the environment: *E. coli*, which transports B$_{12}$ via BtuB and lacks any BtuG homolog, grows readily on 0.4 nM B$_{12}$ (*Di Girolamo et al., 1971*), while *B. thetaiotaomicron* requires BtuG under these conditions (*Figure 1C*). An additional consequence of selection for increased corrinoid binding affinity in BtuG is that such adaptations allow this protein to compete with the host proteins for dietary cobalamin. In this way, the ability of BtuG to acquire B$_{12}$ from IF may have emerged as a byproduct of inter-microbial competition for gut corrinoids. Notably, BtuG has evolved this femtomolar affinity for B$_{12}$ while still maintaining the capacity to release the vitamin for transport into the cell.

Gut bacteria can interfere with host cobalamin absorption in patients with small intestinal overgrowth of bacteria, leading to cobalamin deficiency (*Giannella et al., 1971*). Early efforts to find the responsible bacteria using radiolabeled cyanocobalamin reported that *Bacteroides* were particularly adept at removing cyanocobalamin from IF in vitro (*Giannella et al., 1972*; *Schjönsby et al., 1973*). Other studies describe patients with small intestinal bacterial overgrowth whose cobalamin deficiency was corrected by antibiotics that target *Bacteroides* but not *Proteobacteria* (*Schjönsby et al., 1977*); however, the precise factors responsible for these phenomena were never defined. Our data suggest that BtuG, which is universally present among the *Bacteroides*, could be one extracellular factor responsible for these observations. Although BtuG-mediated vitamin piracy may be well tolerated by a healthy host, conditions of small intestinal bacterial overgrowth or diminished IF production could alter this effect. Notably, even minor B$_{12}$ deficiencies (historically assigned to the 'subclinical' range) could have health consequences (*McCaddon, 2013*; *Moore et al., 2012*).

A comparison of measured binding kinetics of BtuG2 and IF may explain how this bacterial protein extracts cobalamin from IF, which also binds the vitamin very strongly. Reported equilibrium dissociation constants for IF and cobalamin span a broad range ($K_D$ ~10$^{-9}$–10$^{-15}$ M); the most extreme of these describes a $K_D$ ~5 × 10$^{-15}$ M, a $k_{on}$ ~7 × 10$^7$ M$^{-1}$s$^{-1}$ and a $k_{off}$ ~4 × 10$^{-7}$ s$^{-1}$ (*Fedosov et al., 2005*; *Brada et al., 2001*; *Fedosov et al., 2006*). By comparison, BtuG2 has a $K_D$ ~2 × 10$^{-13}$ M, a $k_{on}$ ~1 × 10$^9$ M$^{-1}$s$^{-1}$ and a $k_{off}$ ~3 × 10$^{-4}$ s$^{-1}$ (*Figure 3B*). This $k_{on}$ rate, which is similar to that previously measured in diffusion-limited enzymes and proteins (*Corzo, 2006*), is orders of magnitude greater than the $k_{on}$ for IF. This suggests that BtuG2 has a superior ability to bind up free cobalamin compared with IF.

The different environments in which these proteins encounter free cobalamin could explain these differences in $k_{on}$ rates. In humans, IF first encounters free cobalamin in the duodenum, where the vitamin is released from haptocorrin by host enzymes. At this juncture, there are few gut microbes present (~10$^3$ per gram) to compete with IF for free cobalamin (*Scheithauer et al., 2016*). By contrast, microbial densities in the large intestine exceed 10$^{11}$ per gram, thus introducing a stronger element of competition for free corrinoids among gut microbes (*Whitman et al., 1998*). Therefore,

BtuG2 may face a stronger selective pressure to augment its $k_{on}$ rate in order to remain competitive against other BtuG homologs in the microbiota.

The $k_{off}$ rate of IF is orders of magnitude lower than the $k_{off}$ rate of BtuG2, corresponding to an average protein–ligand association time of ~700 hr for IF–cobalamin versus ~1 hr for BtuG2–cobalamin. Although both of these proteins bind $B_{12}$ for delivery to their respective receptor, their different tasks may explain the observed $k_{off}$ rates. IF binds cobalamin in the proximal small intestine (duodenum), but its uptake receptors on epithelial cells are located exclusively in the distal small intestine (ileum). Therefore, IF must maintain its association with cobalamin while traversing several meters of intestinal tract before the host can absorb its nutrient cargo. By contrast, BtuG2 localizes to the bacterial cell surface and likely acts in cis to capture extracellular corrinoids for subsequent delivery to the outer membrane receptor BtuB on the same cell. This difference in protein function may impose differences in selective pressure for $k_{off}$ rates.

Although BtuG2 and IF thus achieve these strong binding affinities by different kinetics, mixing the proteins results in a transfer of cobalamin from IF to BtuG2 (*Figure 5*). It is unclear whether this occurs through a direct interaction between BtuG2 and IF. Diffusion-limited enzymes and proteins (e.g., acetylcholinesterase and superoxide dismutase) can employ surface electrostatic charges to affect the fluid environment in their immediate vicinity in ways that can enhance the likelihood of protein–ligand contact beyond the frequency determined through diffusion alone (*Tan et al., 1993*; *Getzoff et al., 1983*). Acetylcholinesterase, for example, exhibits contrasting electrostatic charge distributions on opposing faces of the enzyme, creating an electrostatic dipole that is reported to drive the interaction between the enzyme and its positively charged ligand (*Figure 3—figure supplement 2A*) (*Ripoll et al., 1993*; *Tan et al., 1993*). BtuG2 also presents one face with a predominantly positive electrostatic potential and the other with a strikingly negative electrostatic potential (*Figure 3—figure supplement 2A-B*). While the β-propeller structure of BtuG2, which resembles a disk with a central hole, may contribute to the formation of a dipolar electrostatic field through the middle of the protein, these properties are not intrinsic to seven-bladed β-propeller proteins (*Figure 3—figure supplement 2A-B*). Because the coordinated cobalt ion of corrinoids carries a positive charge, surface electrostatic charges could be involved in orienting free corrinoids to facilitate protein–ligand interactions by repelling corrinoids away from the positive electrostatic face while drawing them into the negative electrostatic face of BtuG2. This uncommon surface electrostatic charge distribution in BtuG2 could potentially alter IF–cobalamin stability without direct interaction between the two proteins.

Although our evidence of BtuG lipidation is indirect, the use of cell surface-level machinery to enhance the uptake of key nutrients is not unprecedented for gut microbes. For example, the *Bacteroides* each encode dozens of Sus-like systems, defined by the presence of an outer membrane β-barrel protein (e.g. SusC), a cell surface-exposed lipoprotein (e.g., SusD), and other components that directly interact to digest and import various polysaccharides into the cell (*Koropatkin et al., 2012*; *Glenwright et al., 2017*). Our studies suggest that surface-exposed nutrient binding proteins may determine the ability of these bacteria to not only capture carbon, but also to drive the 'secondary economy' of critical vitamins that power microbial growth in the gut.

# Materials and methods

## Key resources table

| Reagent type (species) or resource | Designation | Source or reference | Identifiers | Additional information |
|---|---|---|---|---|
| Strain, strain background (*E. coli*) | S17-1 lambda pir | PMID_6340113 | | *thi pro hdsR hdsM + recA*, chromosomal insertion of RP4-2(Tc::Mu Km::Tn7), Amp^S |
| Strain, strain background (*E. coli*) | BL21 Rosetta (DE3) | Novagen | | F⁻ ompT hsdS_B(r_B⁻ m_B⁻) *gal dcm* (DE3) pRARE (Cam^R) |

*Continued on next page*

*Continued*

| Reagent type (species) or resource | Designation | Source or reference | Identifiers | Additional information |
|---|---|---|---|---|
| Strain, strain background (*Bacteroides thetaiotaomicron*) | VPI-5482 Δ*tdk* | PMID_18611383 | | |
| Strain, strain background (*B. thetaiotaomicron*) | VPI-5482 Δ*tdk* Δ*locus1* Δ*locus3* | PMID_24439897 | | |
| Strain, strain background (*B. thetaiotaomicron*) | VPI-5482 Δ*tdk* Δ*locus1* Δ*btuG2* Δ*locus3* | This paper | | |
| Strain, strain background (*B. thetaiotaomicron*) | VPI-5482 Δ*tdk* Δ*locus1* Δ*btuB2* Δ*locus3* | PMID_24439897 | | |
| Strain, strain background (*B. thetaiotaomicron*) | VPI-5482 Δ*tdk* Δ*locus1* Δ*locus2* Δ*locus3* | PMID_24439897 | | |
| Strain, strain background (*B. thetaiotaomicron*) | *VPI-5482 Δtdk Δlocus1 Δlocus3 att:: pNBU2_tet_BC01* | This paper | | |
| Strain, strain background (*B. thetaiotaomicron*) | *VPI-5482 Δtdk Δlocus1 ΔbtuG2 Δlocus3 att:: pNBU2_tet_BC14* | This paper | | |
| Strain, strain background (*B. thetaiotaomicron*) | *VPI-5482 Δtdk Δlocus1 ΔbtuB2 Δlocus3 att:: pNBU2_tet_BC14* | This paper | | |
| Strain, strain background (*B. thetaiotaomicron*) | *VPI-5482 Δtdk Δlocus1 ΔbtuG2 Δlocus3 att::pNBU2_tet_BC16 _us1957_btuG2* | This paper | | |
| Strain, strain background (*E. coli*) | *BL21 Rosetta (DE3) pET21_NESG_btuG2* | This paper | | |
| Strain, strain background (*E. coli*) | *BL21 Rosetta (DE3) pET21_NESG_ btuG2_10xHis* | This paper | | |
| Strain, strain background (*E. coli*) | *BL21 Rosetta (DE3) pET21_NESG _BVU2056* | This paper | | |
| Strain, strain background (*E. coli*) | *BL21 Rosetta (DE3) pET21_NESG_ BACUNI04578* | This paper | | |
| Strain, strain background (*E. coli*) | *BL21 Rosetta (DE3) pET21_NESG_ BACCOPRO02032* | This paper | | |
| Recombinant DNA reagent | pExchange-tdk | PMID: 18611383 | | plasmid |
| Recombinant DNA reagent | pExchange_*tdk*_Δ*btuG2* | | | plasmid |
| Recombinant DNA reagent | pNBU2_ermG | PMID: 18611383 | | plasmid |
| Recombinant DNA reagent | pNBU2_ermG_us1957 | This paper | | plasmid |
| Recombinant DNA reagent | pNBU2_tet_BC01 | PMID: 18996345 | | plasmid |
| Recombinant DNA reagent | pNBU2_tet_BC14 | PMID: 18996345 | | plasmid |
| Recombinant DNA reagent | pNBU2_tet_BC16 | PMID: 24439897 | | plasmid |

*Continued on next page*

*Continued*

| Reagent type (species) or resource | Designation | Source or reference | Identifiers | Additional information |
|---|---|---|---|---|
| Recombinant DNA reagent | pNBU2_erm_us1957 _btuG2_C32A | This paper | | plasmid |
| Recombinant DNA reagent | pNBU2_erm_us1957 _btuG2_Δss | This paper | | plasmid |
| Recombinant DNA reagent | pNBU2_erm_us1957 _btuG2_K34A W35A D36A | This paper | | plasmid |
| Recombinant DNA reagent | pNBU2_erm_us1957 _btuG2_susD-ss | This paper | | plasmid |
| Recombinant DNA reagent | pNBU2_erm_us1957 _btuG2_susD-ss-LES | This paper | | plasmid |
| Recombinant DNA reagent | pNBU2_erm_us1957 _btuB2_FLAG_HA | This paper | | plasmid |
| Recombinant DNA reagent | pNBU2_erm_us1957 _BT1763_FLAG_HA | This paper | | plasmid |
| Recombinant DNA reagent | pNBU2_erm_us1957 _BT3704_HA | This paper | | plasmid |
| Recombinant DNA reagent | pET21_NESG_btuG2 | Northeast Structural Genomics Consortium; PDB 3DSM | | plasmid |
| Recombinant DNA reagent | pET21_NESG_btuG2 _10xHis | This paper | | plasmid |
| Recombinant DNA reagent | pET21_NESG_ BVU2056 | This paper | | plasmid |
| Recombinant DNA reagent | pET21_NESG_ BACUNI04578 | This paper | | plasmid |
| Recombinant DNA reagent | pET21_NESG_ BACCOPRO02032 | This paper | | plasmid |

## Bacterial culture conditions

Culturing of *Bacteroides thetaiotaomicron* VPI-5482 was carried out in an anaerobic chamber (Coy Laboratory Products, Grass Lake, MI, USA), filled with 70% $N_2$, 20% $CO_2$, and 10% $H_2$ by volume, using minimal media with vitamin $B_{12}$ omitted (*Martens et al., 2008*) supplemented where specified with 500 µM DL-methionine and/or vitamin $B_{12}$ (0, 0.37 or 37 nM). *Escherichia coli* S17-1 lambda *pir* or BL21 Rosetta (DE3) strains were grown in LB medium and incubated aerobically at 37°C. Culture media were supplemented with antibiotics as needed at the following concentrations: ampicillin 100 µg/mL, chloramphenicol 30 µg/mL, erythromycin 25 µg/mL, gentamicin 200 µg/mL, tetracycline 2 µg/mL, and 5-fluoro-2′-deoxyuridine (FUdR) 200 µg/mL.

## Genetic techniques

Plasmid constructs (*Supplementary file 1*) were created, maintained and transformed using standard molecular cloning procedures. Primers (*Supplementary file 1*) were obtained from the Keck Biotechnology Resource Laboratory (Yale University, New Haven, CT, USA) and DNA amplification was performed using KAPA HiFi ReadyMix (Kapa Biosystems, Wilmington, MA, USA). Gene deletions in *B. thetaiotaomicron* were carried out as previously reported using *B. thetaiotaomicron* Δ*tdk* Δ*locus1* Δ*locus3* as a parent strain (*Degnan et al., 2014a*) by amplifying flanking regions (~1000 bp) of genes of interest and joining them by splicing by overlap extension (SOE) PCR or Gibson assembly. The concatenated fragments were inserted into the suicide vector pExchange-tdk (*Martens et al., 2008*) via ligation or Gibson assembly. Clones were sequence-verified and introduced into *B. thetaiotaomicron* Δ*tdk* Δ*locus1* Δ*locus3* by conjugation. Following counter selection, gene deletions were confirmed by PCR. Gene complementation constructs were created using pNBU2 vectors (with or without oligonucleotide barcodes) introduced in single copy into *B. thetaiotaomicron* as previously described (*Martens et al., 2008*). Complementation constructs contained

425 bp upstream of BT1957, the first gene in *locus2*, to capture the *locus2* promoter and vitamin B$_{12}$ riboswitch (*us1957*, *Supplementary file 1*).

## Comparative genomics and computational analyses

### Identification of BtuG homologs

BlastP searches were carried out with BT1490, BT1954 and BT2095 as queries against the predicted proteomes of 313 human gut bacteria (*Degnan et al., 2014a*) and e-value cutoff of 1e-10. Results were collated, de-replicated and BtuG homologs assigned to *btuB*-containing operons described previously (*Degnan et al., 2014a*). *btuB*-containing operons without a *btuG* homolog and *btuG*-containing operons without a previously characterized *btuB* gene were manually inspected using BlastP and Phyre2 (*Altschul et al., 1990*; *Kelley and Sternberg, 2009*).

### BtuG sequence logo

The first 90 amino acids of 114 BtuG homologs (*Supplementary file 1*) were aligned in ClustalW (*Larkin et al., 2007*). Their sequence conservation was displayed via sequence logo (*Crooks et al., 2004*; *Schneider and Stephens, 1990*).

### BtuG gene tree

A multiple sequence alignment of 114 homologs of BtuG2 (*Supplementary file 1*) was generated in MAFFT (*Katoh et al., 2002*) and converted into a gene tree using FigTree.

### Protein surface electrostatic analysis

PDB 3DSM, 3S25, 1L0Q and 1GQR were converted to PQR files using the PDB2PQR server under default settings (*Dolinsky et al., 2004*). The resulting surface electrostatic profiles were then analyzed and viewed using an adaptive Poisson-Boltzmann solver (APBS) in PyMOL.

### Protein structure overlay

The crystal structures of BtuG2 (PDB 3DSM) and the N-terminal domain of the *M. mazei* S-layer protein MSMAS_RS11935 (PDB 1L0Q, Chain A; *Jing et al., 2002*) were aligned by super-positioning of the C-alpha backbones using the TOPP program in the CCP4 protein crystallography suite (*Lu, 1996*), Collaborative Computational Project Number 4, 1994). Images were prepared using Chimera (*Pettersen et al., 2004*).

## Growth curves

Overnight cultures of *B. thetaiotaomicron* strains grown in minimal medium supplemented with methionine (no B$_{12}$) were pelleted, washed three times in minimal medium without methionine or vitamin B$_{12}$, and used to inoculate wells of a 96-well plate in triplicate containing minimal media with methionine or vitamin B$_{12}$ (0, 0.37 or 37 nM). The plate was incubated anaerobically under constant agitation for 72 hr at 37°C and OD600 measurements were taken at regular intervals using a BioTek Eon microplate spectrophotometer.

## In vitro bacterial competitions

*B. thetaiotaomicron* strains carrying unique oligonucleotide barcodes were co-cultured in minimal media in triplicate as previously described (*Degnan et al., 2014a*; *Martens et al., 2008*). Briefly, *B. thetaiotaomicron* strains were grown overnight in minimal media supplemented with methionine, washed and resuspended in minimal media without methionine or vitamin B$_{12}$. OD600 was measured and used to create a 1:10 mixture of competing strains (1 part parent strain to 10 parts Δ*btuG2* strain), which was then used to inoculate, at 1:1000, minimal media supplemented with methionine or vitamin B$_{12}$ (0, 0.37, or 37 nM). These inoculations were then incubated anaerobically at 37 °C under static conditions or shaking (250 rpm). Cultures were passaged at 1:1000 into fresh media every 24 hr and an aliquot was stored at −20 °C for gDNA extraction (*Truett et al., 2000*). Relative strain abundances were determined by quantitative PCR (qPCR) using a CFX96 thermocycler (Bio-Rad, Hercules, CA, USA) and SYBR FAST Universal Mastermix (KAPA Biosystems, Wilmington, MA, USA) (*Degnan et al., 2014a*). Strain abundances were analyzed using a standard curve and efficiency-corrected ΔCq method was used to determine relative fold changes (*Bookout et al., 2006*).

## Gnotobiotic animal studies

All animal experiments were performed using protocols approved by the Yale University Institutional Animal Care and Use Committee. Male and female germfree 8- to 12-week-old Swiss Webster mice were individually caged and maintained in flexible plastic gnotobiotic isolators with a 12 hr light/dark cycle. Mice were provided with standard autoclaved mouse chow (5K67 LabDiet; Purina, St. Louis, MO, USA) *ad libitum*. Germfree mice were colonized with 200 µL bacterial glycerol stocks by oral gavage. Mice were divided into groups (*n* = 4 – 5/group). Each mouse in the first group was gavaged with $10^8$ CFU each of *B. thetaiotaomicron Δtdk Δlocus1 Δlocus3 att1*::pNBU2_tetQ_BC01 ('parent' strain) and *B. thetaiotaomicron Δtdk Δlocus1 Δlocus3 ΔbtuG2 att1*::pNBU2_tetQ_BC14 ('*ΔbtuG2*' strain). Each mouse in the second group was gavaged with $10^8$ CFU each of the parent strain and *B. thetaiotaomicron Δtdk Δlocus1 Δlocus3 ΔbtuB2 att1*::pNBU2_tetQ_BC14 ('*ΔbtuB2*' strain). Each mouse in the third group was gavaged with ~$10^7$ CFU each of the parent strain, the *ΔbtuG2* strain, and *B. thetaiotaomicron Δtdk Δlocus1 Δlocus3 ΔbtuG2 att1*::pNBU2_tetQ_B-C16_us1957_btuG2 ('*ΔbtuG2 + btuG2*' strain). Fecal samples were collected over time and stored at −80°C before genomic DNA extraction. DNA was extracted as described previously (*Cullen et al., 2015*). The relative abundance of each strain was determined using oligonucleotide barcode-specific primers (*Supplementaryfile 1*) in a qPCR assay as described above.

## $B_{12}$ riboswitch biosensor assays

*B. thetaiotaomicron* strains were grown in triplicate anaerobically at 37 °C to mid-log phase (OD600 ~0.3) in minimal media supplemented with methionine and vitamin $B_{12}$ (0, 0.37, or 37 nM). RNA was extracted using a cell lysis buffer (10 mM Tris pH 8.0, 1 mM EDTA, 0.2 mg lysozyme, 0.5 mg proteinase K) and an RNeasy kit (Qiagen, Hilden, Germany). DNA was removed using DNA-free DNA Removal Kit (Invitrogen, Carlsbad, CA, USA), and RNA was again cleaned using an RNeasy kit (Qiagen, Hilden, Germany). cDNA was then made with SuperScript II Reverse Transcriptase (Invitrogen, Carlsbad, CA, USA) using the manufacturer's instructions, and RNA was removed with 1 N NaOH at 65 °C for 30 min and neutralized with 1 N HCl. Samples were then cleaned using a PCR purification kit (Qiagen) and cDNA was quantified using a Qubit (Invitrogen, Carlsbad, CA, USA). Quantitative PCR was performed using SYBR FAST Universal Mastermix (KAPA Biosystems, Wilmington, MA, USA) and gene-specific primers (*Supplementary file 1*). Samples were normalized first to 16S rRNA expression for each individual sample and replicate, and then normalized to the expression level of each strain in 0 nM $B_{12}$ (*Figure 1D*) or the *btuG2* complement strain in 0.37 nM $B_{12}$ (*Figure 2—figure supplement 1B*). A standard curve and efficiency-corrected ΔCq method was used to determine relative fold changes (*Bookout et al., 2006*).

## Immunoblotting

Detection of BtuG2 from *B. thetaiotaomicron* lysates was performed by Western blot analysis using a custom-made rabbit anti-BtuG2 polyclonal antibody (Cocalico Biologicals, Reamstown, PA, USA).

## Proteinase K assay

A *B. thetaiotaomicron* strain expressing an HA-epitope tagged allele of the periplasmic protein SusA (*Shipman et al., 1999*) was grown to OD600 ~0.8 in minimal media with methionine, pelleted and washed in 1x cOmplete EDTA-free protease-inhibitor cocktail (Roche, Basel, Switzerland) before being pelleted and stored at −80 °C. Pellets were thawed and resuspended in PBS with proteinase K (0, 10, 50 or 100 µg/mL; AmericanBio, Natick, MA, USA), and incubated at 37 °C aerobically under continuous agitation (250 rpm) for 8 hr. Cells were then pelleted and washed 3 times in 1x cOmplete EDTA-free protease-inhibitor cocktail, pelleted and stored at −80 °C. Thawed cells were lysed using BugBuster reagent (Millipore Sigma, Burlington, MA, USA), 20 µg of clarified protein lysate was loaded onto an SDS-PAGE gel, transferred to a PVDF membrane and probed with rabbit anti-BtuG2 and rabbit anti-HA (Santa Cruz Biotechnology, Dallas, TX, USA).

## Cell fractionation

*B. thetaiotaomicron* cultures were grown to OD600 ~0.6 in minimal media with methionine. Cells were pelleted (~3000 x g for 15 min at 4°C) and supernatant was filtered through 0.2 µm filter and stored temporarily on ice. Pellets were resuspended in breakage buffer (50 mM Tris pH 8, 5 mM

EDTA, 2 mM PMSF, 10% glycerol), lysed at 4°C by sonication (40 Amps; 15 s 'on' and 30 s 'off'; 3 min total), and clarified lysates were ultracentrifuged at 100,000 x g for 1 hr at 4°C to separate membranes (insoluble) from cytoplasm/periplasm (soluble) fractions. Membrane fractions were resuspended in 250 μl of breakage buffer, while the cytoplasm/periplasm fraction was concentrated by centrifugal filtration (30K; Millipore Sigma, Burlington, MA, USA) to 250 μl. Membrane and cytoplasm/periplasm fractions were temporarily stored on ice. Filtered supernatant was utracentrifuged at ~100,000 x g for 1 hr at 4°C to remove outer membrane vesicles. The soluble fraction was then concentrated by centrifugal filtration to 250 μl. 20 μl each of membrane, cytoplasm/periplasm, and supernatant fractions were loaded onto SDS-PAGE gels and analyzed by Western blot. PVDF membranes were probed with rabbit anti-BtuG2 and mouse anti-RpoB (Santa Cruz Biotechnology, Dallas, TX, USA) as a cytoplasmic control.

## Co-immunoprecipitation

*B. thetaiotaomicron* strains were grown to OD600 ~0.6 in minimal media with methionine. Cells were pelleted, supernatant was removed, and cells were lysed with BugBuster reagent. Co-immunoprecipitation was carried out using FLAG HA Tandem Affinity Purification kit (Millipore Sigma, Burlington, MA, USA) according to the manufacturer's instructions. Eluates were probed with rabbit anti-BtuG2 by Western blot.

## *Trans*-complementation assays

Overnight cultures of *B. thetaiotaomicron* strains were grown in minimal media with methionine. Supernatant donor strains were subcultured (1:100) into 60 mL fresh minimal media with methionine and allowed to grow to OD600 ~0.6. Supernatant recipient strains were washed 3 times in minimal media without methionine or vitamin $B_{12}$, subcultured to a final OD600 ~0.001 in 1 mL minimal media without methionine or vitamin $B_{12}$, and incubated at 37 °C anaerobically for 8 – 12 hr. Supernatant donor cultures were pelleted and supernatant filtered (0.2 μm). Filtered supernatants were then ultracentrifuged at 100,000 x g for 1 hr at 4 °C to remove outer membrane vesicles and concentrated to <2 mL by centrifugal filtration (30K; Millipore Sigma, Burlington, MA, USA). Where indicated, concentrated supernatants were supplemented with 0.37 μM vitamin $B_{12}$ or PBS and incubated for 20 min at room temperature aerobically on a nutator. Supernatants were then washed 4 times in 70 mL minimal media without methionine or vitamin $B_{12}$ by centrifugal filtration (30K). Washed, concentrated supernatants were then applied to recipient cell cultures and incubated at 37 °C anaerobically. CFU measurements were taken at regular intervals. For assays involving IF, recombinant human IF (Xeragenx LLC) was incubated in 0.37 μM vitamin $B_{12}$ or PBS for 20 min at room temperature on a nutator, and washed 4 times in 70 mL minimal media without methionine or vitamin $B_{12}$ via centrifugal filtration (30K). IF (± vitamin $B_{12}$) was provided either alone to recipient cell cultures (e.g., *Figure 5A*) or with $B_{12}$-free donor supernatants (e.g., *Figure 5C*) at a final concentration of 10 nM IF per replicate. Cultures were incubated at 37°C anaerobically and CFU measurements were taken at regular intervals.

## Size-exclusion chromatography with multi-angle light scattering (SEC-MALS)

Recombinant BtuG2-6xHis or BtuG2-10xHis was expressed and purified from *E. coli* BL21 Rosetta (DE3) carrying a modified pET21 vector. Cells were grown to OD600 ~0.6 before being induced for 3 hr in 0.5 mM IPTG at 37 °C under constant agitation. Cell pellets were lysed using BugBuster reagent (Millipore Sigma, Burlington, MA, USA). Lysates were incubated for 1 hr at 4 °C with Ni-NTA agarose beads (Qiagen, Hilden, Germany) and washed with 12 – 18 mL of wash buffer (50 mM $NaH_2PO_4$, 300 mM NaCl, 20 mM imidazole pH 7.4), and eluted with 6 mL elution buffer (50 mM $NaH_2PO_4$, 300 mM NaCl, 250 mM imidazole, pH 7.4). BtuG2-6xHis or BtuG2-10xHis fractions were dialyzed overnight in 20 mM Tris pH eight before being spun through a Pierce™ strong anion exchange column (ThermoFisher Scientific, Waltham, MA, USA) according to the manufacturer's instructions. Eluted proteins were dialyzed twice for 4–8 hr in 4 L of PBS, pH 8. Proteins were quantified by Bradford assay (Bio-Rad, Hercules, CA, USA) according to the manufacturer's instructions. BtuG2-6xHis was incubated at a 1:1 molar ratio of cyanocobalamin to protein for 30 min at 25 °C and analyzed by SEC-MALS. BtuG2-10xHis was enriched for protein monomers by size exclusion

chromatography and quantified using a NanoDrop spectrophotometer (ThermoFisher Scientific, Waltham, MA, USA) for use in SPR experiments (described below).

## Cyanocobalamin binding by BtuG homologs from *B. thetaiotaomicron*, *B. vulgatus*, *B. uniformis* and *B. coprophilus*

pET21_NESG was used to express and purify C-terminal 6xHis-tagged versions of the BtuG homologs BVU2056, BACUNI04578 and BACCOPRO02032 as described above for BtuG2. Each BtuG homolog, or BSA or PBS as controls, was incubated at a 1:1 molar ratio of cyanocobalamin to protein for 30 min at 25 °C before being spun through a centrifugal filter (30K; Millipore Sigma, Burlington, MA, USA) to elute unbound cyanocobalamin. Proteins were washed 3 times with 400 µl PBS. Retained protein-cyanocobalamin complexes were resuspended in 100 µl PBS and analyzed by spectrophotometry for absorbance at 360 nm, corresponding to cyanocobalamin.

## Surface plasmon resonance (SPR)

Binding studies were performed at 25°C using a Biacore T100 optical biosensor (GE HealthCare, Biacore, Piscataway, NJ, USA). Recombinant BtuG2-10xHis was purified from *E. coli* BL21 Rosetta (DE3) as described above and immobilized on a NTA chip. Cyanocobalamin was injected at 0.8, 0.4, 0.2, 0.1, and 0.05 nM and the binding was monitored in single cycle kinetics in PBS (*Karlsson et al., 2006*). Binding responses were double-referenced against non-specific binding to dextran and the NTA surface alone, and against injections of buffer alone. Binding affinity was determined by fitting the kinetics of the binding reaction to a 1:1 binding model using BioEvaluation software (GE HealthCare, Biacore, Piscataway, NJ, USA).

# Acknowledgements

We thank members of ALG's laboratory; Michiko Taga, Kenny Mok, and Nicole Koropatkin for helpful discussions; the Northeast Structural Genomics Consortium for pET21_NESG_BT1954; Robert Doyle and Xeragenx LLC for the gift of recombinant human intrinsic factor. This work was supported by NIH grants GM118159, GM103574, the Burroughs Wellcome Fund, and the Howard Hughes Medical Institute Faculty Scholars Program to ALG, and support from the Gruber Science Fellowship Program to AGW.

# Additional information

## Funding

| Funder | Grant reference number | Author |
| --- | --- | --- |
| National Institutes of Health | GM118159 | Andrew L Goodman |
| Burroughs Wellcome Fund | 1014860 | Andrew L Goodman |
| Howard Hughes Medical Institute | 55108526 | Andrew L Goodman |
| Gruber Foundation | Graduate Student Fellowship | Aaron G Wexler |
| National Institutes of Health | GM103574 | Andrew L Goodman |

The funders had no role in study design, data collection and interpretation, or the decision to submit the work for publication.

## Author contributions

Aaron G Wexler, Data curation, Formal analysis, Investigation, Visualization, Methodology, Writing—original draft; Whitman B Schofield, Investigation, Writing—review and editing; Patrick H Degnan, Formal analysis, Investigation, Methodology, Writing—review and editing; Ewa Folta-Stogniew, Formal analysis, Methodology; Natasha A Barry, Investigation, Methodology; Andrew L Goodman, Conceptualization, Resources, Formal analysis, Supervision, Funding acquisition, Writing—original draft, Writing—review and editing

## Author ORCIDs

Aaron G Wexler (iD) http://orcid.org/0000-0001-7926-089X
Andrew L Goodman (iD) http://orcid.org/0000-0001-7599-3471

## Ethics

Animal experimentation: This study was performed in strict accordance with the recommendations in the Guide for the Care and Use of Laboratory Animals of the National Institutes of Health. All of the animals were handled according to approved institutional animal care and use committee (IACUC) protocols (#2017-11423) of Yale University.

## Decision letter and Author response

Decision letter https://doi.org/10.7554/eLife.37138.015
Author response https://doi.org/10.7554/eLife.37138.016

## Additional files

### Supplementary files

• Supplementary file 1. (Table S1) Identified BtuB-BtuG pairs from BlastP and Phyre2 searches of BT1490, BT1954 and BT2095. (Table S2) Amino acid sequences of 114 BtuG homologs used to create the sequence logo in *Figure 2A*. (Table S3) Bacterial strains, plasmids and oligonucleotide primers used in this study.
DOI: https://doi.org/10.7554/eLife.37138.012

• Transparent reporting form
DOI: https://doi.org/10.7554/eLife.37138.013

All data generated or analyzed during this study are included in the manuscript and supporting files. Source data files have been provided for Figures 1A and 2A.

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
