## [Decision Letter]

Thank you for submitting your article "Human Gut Bacteria Capture Vitamin B_12_ from their Environment and Host via Cell Surface-Exposed Lipoproteins" for consideration by *eLife*. Your article has been reviewed by three peer reviewers, and the evaluation has been overseen by a Reviewing Editor and Wendy Garrett as the Senior Editor. The following individual involved in review of your submission has agreed to reveal his identity: Jan Peter van Pijkeren (Reviewer #3).

The reviewers have discussed the reviews with one another and the Reviewing Editor has drafted this decision to help you prepare a revised submission.

Summary:

The reviewers agreed that this paper represents a very interesting, well-designed, and well-conducted study of vitamin B_12_ uptake in an important gut commensal. The authors center their study on BtuB, an outer membrane lipoprotein from *Bacteroides thetaiotaomicron*. They perform a series of careful biochemical analyses that reveal that BtuG is essential for B_12_ transport into the cell. They further show that BtuG is key for the in vivo fitness of *B. thetaiotaomicron*. A major claim is that BtuG is involved in "vitamin piracy" by competing with human intrinsic factor for vitamin B_12_ binding.

Overall, the study presents novel insight into how gut commensals obtain essential vitamins in the complex gastrointestinal environment. The study is well-conducted, well-written, and the data on BtuG function in *B. thetaiotaomicron* vitamin B_12_ uptake is convincing. The reviewers felt that overall, the work provides valuable new insight into nutrient acquisition strategies of intestinal commensal bacteria. However, as detailed below, there were concerns about the basis for the vitamin piracy claim that need to be addressed in a revised manuscript.

Essential revisions:

1) A major claim of this study is that BtuG is involved in "vitamin piracy" by capturing vitamin B_12_ from human intrinsic factor. The experiments supporting this claim were all conducted in vitro. In vivo experiments showing that the wild-type bacteria capture B_12_ from the host, while the mutant does not, would greatly strengthen this central claim of the paper. However, the reviewers recognized that such an in vivo experiment could pose some difficult technical challenges and might not be doable in a restricted time-frame. In that case, we suggest that the title and Abstract be revised with less emphasis on the idea of B_12_ capture from the host. In addition, the discussion would need to be improved by discussing alternative possibilities as suggested in point 3 below.

2) *∆btuG2* and *∆btuB2* are outcompeted by the parent strain in a germ-free mouse (Figure 1E, F). Since this is the only in vivo data demonstrating the role of these proteins in intestinal fitness, it is important that – at least for one of these proteins – that its function is complemented to conclusively demonstrate that the reduction in fitness is attributable to deletion of *btuG2* or *btuB2*.

3) Vitamin B_12_ availability in the colon is an essential underpinning of the vitamin piracy claim. In the discussion amongst the reviewers, there were questions raised about whether vitamin B_12_ is likely to be freely available in the colon given that it is typically absorbed in the small intestine. This may also be a point of confusion for some readers, and thus we suggest some further discussion and/or clarification of this point. The following points were raised as possible avenues for clarification:

a) A reviewer pointed out that there can be small intestinal populations of *B. theta* (albeit at lower density) – is it possible that the piracy is therefore occurring primarily among these small intestinal populations rather than those present in the colon?

b) Is it possible that BtuG could be a factor that has evolved to allow successful competition with other microbes, rather than with the host? This would plausibly constitute enough selection pressure for the high affinity of the protein for B_12_ to emerge.

c) Can *B. thetaiotaomicron* utilize pseudovitamin B_12_, and – if so – how does the utilization (i.e. growth) compare to vitamin B_12_? Pseudovitamin B_12_ is produced by select gut bacteria (i.e. *Lactobacillus reuteri* and *Eubacterium hallii*), food-associated microbes (i.e. *Arthrospira platensis*), or used as a food additive. Although pseudovitamin B_12_ is another source of B_12_, it cannot be absorbed in the human GI tract (it does bind IF). If *B. thetaiotaomicron* can utilize pseudovitamin B_12_ more efficiently than B_12_, then *B. thetaiotaomicron* would be beneficial for human health rather than detrimental, as the authors speculate in their Discussion (third paragraph).

---

## [Author Response]

Essential Revisions:1) A major claim of this study is that BtuG is involved in "vitamin piracy" by capturing vitamin B_12_ from human intrinsic factor. The experiments supporting this claim were all conducted in vitro. In vivo experiments showing that the wild-type bacteria capture B_12_ from the host, while the mutant does not, would greatly strengthen this central claim of the paper. However, the reviewers recognized that such an in vivo experiment could pose some difficult technical challenges and might not be doable in a restricted time-frame. In that case, we suggest that the title and Abstract be revised with less emphasis on the idea of B_12_ capture from the host. In addition, the discussion would need to be improved by discussing alternative possibilities as suggested in point 3 below.

We agree with this suggestion and have revised the title, Abstract, and Discussion accordingly.

2) ∆btuG2 and ∆btuB2 are outcompeted by the parent strain in a germ-free mouse (Figure 1E, F). Since this is the only in vivo data demonstrating the role of these proteins in intestinal fitness, it is important that – at least for one of these proteins – that its function is complemented to conclusively demonstrate that the reduction in fitness is attributable to deletion of btuG2 or btuB2.

It has been previously shown that the competitive fitness defect of the *btuB2* mutant in vivo can be complemented by expression of the gene in single copy from a heterologous chromosomal locus (Degnan et al., 2014, Figure 4A, B). We have added this information to the paper.

In response to this comment, we conducted new experiments measuring the fitness of wildtype, *∆btuG2*, and complemented strains in gnotobiotic mice. These new data have been added to the paper as Figure 1—figure supplement 1.

3) Vitamin B_12_ availability in the colon is an essential underpinning of the vitamin piracy claim. In the discussion amongst the reviewers, there were questions raised about whether vitamin B_12_ is likely to be freely available in the colon given that it is typically absorbed in the small intestine. This may also be a point of confusion for some readers, and thus we suggest some further discussion and/or clarification of this point. The following points were raised as possible avenues for clarification:a) A reviewer pointed out that there can be small intestinal populations of B. theta (albeit at lower density) – is it possible that the piracy is therefore occurring primarily among these small intestinal populations rather than those present in the colon?

We agree with the reviewer and have suggested this possibility (see subsection “*B. thetaiotaomicron* can use BtuG2 to acquire cyanocobalamin from human intrinsic factor” and Discussion, paragraph six).

b) Is it possible that BtuG could be a factor that has evolved to allow successful competition with other microbes, rather than with the host? This would plausibly constitute enough selection pressure for the high affinity of the protein for B_12_ to emerge.

Yes, we think this is the most likely explanation for our observations (see Discussion, paragraph two).

c) Can B. thetaiotaomicron utilize pseudovitamin B_12_, and – if so – how does the utilization (i.e. growth) compare to vitamin B_12_? Pseudovitamin B_12_ is produced by select gut bacteria (i.e. Lactobacillus reuteri and Eubacterium hallii), food-associated microbes (i.e. Arthrospira platensis), or used as a food additive. Although pseudovitamin B_12_ is another source of B_12_, it cannot be absorbed in the human GI tract (it does bind IF). If B. thetaiotaomicron can utilize pseudovitamin B_12_ more efficiently than B_12_, then B. thetaiotaomicron would be beneficial for human health rather than detrimental, as the authors speculate in their Discussion (third paragraph).

We appreciate this useful point. We have previously shown that *B. thetaiotaomicron* can utilize [Ade]Cba (pseudovitamin B_12_); growth rates in minimal medium lacking methionine and supplemented with either [Ade]Cba or cyanocobalamin are equivalent (Degnan et al., 2014, Figure S3A). Others have reported that the binding affinity of IF to [Ade]Cba is an order of magnitude weaker than IF to B_12_ (Fedozov et al., Eur J Biochem, 2003; Stupperich and Nexo, Eur J Biochem, 1991).

We are reluctant to speculate about the interactions between BtuG, IF and [Ade]Cba because we have not studied the interactions between these proteins and [Ade]Cba. We think that it is unlikely that BtuG2 acts as a “filter” to distinguish corrinoids, because it binds B_12_ and cobinamide with similar affinity (Figure 3B). Thus, future studies will need to examine binding of diverse corrinoids (including those not commercially available) with many proteins in the corrinoid transport loci in addition to BtuG.